# Multi-Input Ćuk-Derived Buck-Boost Voltage Source Inverter for Photovoltaic Systems in Microgrid Applications

**Eltaib Abdeen [1], Mahmoud A. Gaafar [1], Mohamed Orabi [1,*], Emad M. Ahmed [1,2] and Abdelali El Aroudi [3,*]**

[1] APEARC, Faculty of Engineering, Aswan University, Aswan 81542, Egypt; eabdeen@apearc.aswu.edu.eg (E.A.); mgaafar@apearc.aswu.edu.eg (M.A.G.); eelbakoury@apearc.aswu.edu.eg (E.M.A.)

[2] Department of Electrical Engineering, Faculty of Engineering, Jouf University, Sakaka 2014, Saudi Arabia

[3] Departament d'Enginyeria Electronica, Electrica I Automatica, Universitat Rovira i Virgili, Av. Paisos Catalans, no. 26, 43007 Tarragona, Spain

\* Correspondence: morabi@apearc.aswu.edu.eg (M.O.); abdelali.elaroudi@urv.cat (A.E.A.)

**Abstract:** This paper presents a multi-input Ćuk-derived Buck-Boost voltage source inverter (CBBVSI) for Photovoltaic (PV) systems. The proposed topology consists of a single-stage DC-AC inverter that combines both DC-DC and DC-AC stages. The DC-DC stage is used for stepping-up the voltage from the PV generator. Simultaneously, the DC-AC stage is used for interfacing the PV source with the AC grid. The topology allows three sources to utilize the antiparallel diodes for each inverter leg for transferring the energy. The proposed system exhibits several features such as a reduction of the number of components compared to typical two-stage structures, and Split-Source Inverter (SSI), and Z-Source Inverter (ZSI) topologies. Moreover, the power of each PV source can be harvested either simultaneously or separately since independent Maximum Power Point Tracking (MPPT) is performed. The system was simulated using MATLAB/SIMULINK software and a 1 kW laboratory prototype was implemented to verify the operation of the proposed CBBVSI. The numerical simulations are presented together with the experimental results, showing a good agreement.

**Keywords:** PV; buck-boost inverter; multi-input inverters; MPPT

## 1. Introduction

The use of Photovoltaic (PV) systems has recently grown in both residential and industrial applications. The most important part in a PV system is the power–electronic interface. Therefore, the main challenge for the correct operation of a PV system is the design of this interface. Two main categories of power–electronic interfaces from the power stage point of view can be recognized. These are single-stage and two-stage inverters. The latter can be implemented by cascading two stages. The first one is a DC-DC converter that has the function of stepping up the DC voltage to match the grid voltage, while extracting the maximum power from the PV source. The second stage is a DC-AC inverter that has the function of either injecting the AC current into the grid [1,2] or to directly supply the load [3]. A power decoupling capacitor is used between the first stage and the second stage. The problem of the two-stage approach is the large size and high cost due to the increased number of components. Therefore, one of the main challenging tasks in PV systems is single-stage inverter implementation with a reduced size and cost [4,5].

A number of single-stage topologies have been proposed in [6–9]. In all of these proposed topologies no boost conversion stage has been used, meaning that either a large number of PV modules

must be series-connected to obtain a PV voltage higher than the grid voltage, or a low-frequency transformer must be used. Using a large number of series-connected PV modules adds issues due to either shadowing effects caused by the possible existence of clouds, trees, or buildings, or due to module manufactural mismatch. To overcome this challenge, a DC-DC boost converter combined with a DC-AC inverter is usually used. In [10], a single-stage single-phase operation has been achieved by increasing the input voltage range. However, additional semiconductor switches and a large DC link capacitor have been used, increasing the total cost and decreasing the lifetime. The Z-source inverter (ZSI) topology has been proposed to merge the DC-DC converter with a DC-AC inverter [11]. ZSI can step-up the input source voltage and can perform the power conversion in a single stage. However, large passive elements must be used and the duty cycle operation range is limited [11–13], leading to a large size and low boosting gain. To overcome the problem of the low conversion gain, switched inductor ZSI and Qazi Z-source inverter (qZSI) have been used in [14–17]. However, additional passive elements at the impedance network have been required, which increases the size and cost of the topology. Y-source and quasi-Y source inverters have been considered in [18–20]. The inverter can provide high boosting gain and modulation index but the coupled transformer with three windings adds more size and cost. Based on a full-bridge single-phase inverter, a single-stage DC-AC for PV applications has been presented in [21], where one inductor and two diodes have been added to step-up the input voltage. Using the same idea in [21], a Split Source Inverter (SSI) three-phase inverter has been considered in [22]. Compared to ZSI, a Split Source Inverter (SSI) circuit has some advantages, such as the reduction in the number of passive components and voltage ratios. However, this topology uses three diodes and adds more conduction losses, reducing the overall efficiency. Moreover, the input current exhibits low-frequency oscillations caused by the variations of the duty cycle. An improvement was proposed in [23] on SSI topology, where an inductor connected to two MOSFETs, and a PWM with the simultaneous use of constant and sinusoidal duty cycles on two legs of the full-bridge inverter have been used. However, in the presented topology, three switches instead of three diodes and a large capacitor are used. Recently, another modification of the topology presented in [23] was proposed in [24] by removing one active switch. In that work, it was reported that an enhanced voltage boosting gain can be obtained. In [25] a Buck-Boost Voltage Source Inverter (BBVSI), derived from the Ćuk and SEPIC topologies, has been presented. The analysis of SEPIC-derived BBVSI (SBBVSI) has been presented in [26]. One of the merits of the single-stage Ćuk-derived BBVSI (CBBVSI), shown in Figure 1, when compared to the ZSI, is the use of smaller passive components. The analysis of BBVSI operation for a single input was proposed in [27]. However, a consistent performance evaluation of the Ćuk-derived BBVSI is still missing and requires an investigation of the system behavior, especially when operated as a multi-input inverter.

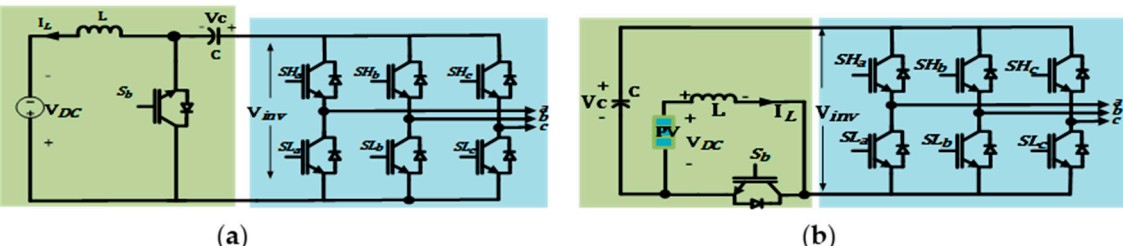

**Figure 1.** Ćuk-derived buck-boost inverter; (**a**) the circuit of [25], (**b**) re-arranged Ćuk-derived buck-boost inverter.

A family of multi-input converters based on three switches leg has been presented in [28], where one of the inputs should be a battery and the other ones could be PV generators with either buck mode or boost mode operation, depending on the status of the battery. One drawback of this approach is a reduced overall conversion gain. The main contribution of this paper is introducing and analyzing single-stage three-phase multi-input CBBVSI for distributed PV systems that can be used in microgrid

applications (Figure 1). All of the inputs to the system could be PV modules and each PV source has its own MPPT.

The rest of this paper is organized as follows. The operation of single-input CBBVSI and the proposed multi-input CBBVSI are presented in Section 2. Section 3 deals with the modulation of the system and determination of its conversion ratio. The devices stress is also studied in the same section. The control of the proposed multi-input inverter is discussed in Section 4. To verify the operation of the single-input CBBVSI and the proposed multi-input inverter, simulation and experimental tests for different conditions are performed in Sections 5 and 6, respectively. In Section 7, a comparison is presented between the proposed and existing topologies. Finally, the conclusions of this study are drawn in Section 8.

## 2. The Operation Principle of the Proposed Topology

### 2.1. The Operation of the Single-Input CBBVSI Topology

The schematic diagram for the inverter considered in [23] is shown in Figure 1a and then rearranged, as in Figure 1b. It uses the same structure of the common VSI. The step-up voltage is achieved using one switch $S_b$. This switch is turned ON with duty cycle $D_b$ based on MPPT control for any state of the inverter. Two modes of the operation are employed.

**Mode (1), or inductor charging mode:** During this mode, $S_b$ is switched ON, the inductor $L$ is charged from the PV source and the capacitor $C$ feeds the AC load through the switches ($SH_{a,b,c}$ and $SL_{a,b,c}$). The inverter voltage $V_{inv}$ is equal to the capacitor voltage $V_C$, as shown in Figure 2a. By applying the small ripple approximation, the equations describing the behavior of the inverter during this mode, can be expressed as follows:

$$V_L = L\frac{di_L}{dt} = V_{dc} \tag{1}$$

$$V_{inv} = V_c \tag{2}$$

where $V_{dc}$ is the DC input voltage, and $V_C$ is the capacitor voltage.

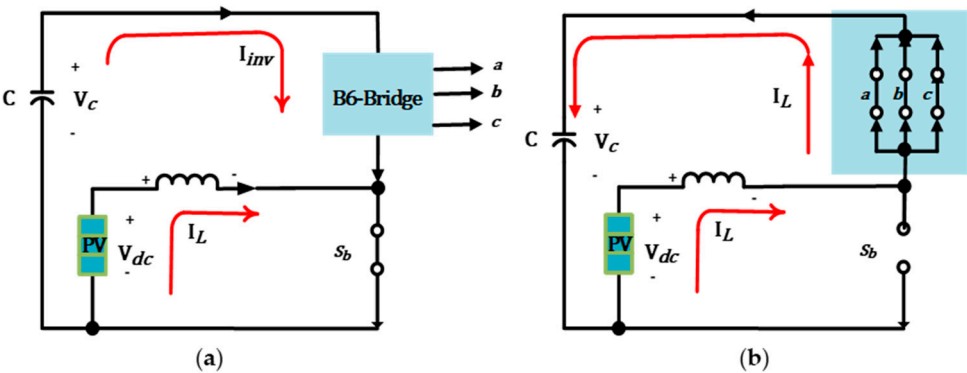

**Figure 2.** The simplified equivalent schematic circuit diagram of the single-input Ćuk-derived Buck-Boost voltage source inverter (CBBVSI); (**a**) Mode 1: charging mode; (**b**) Mode2: discharging mode.

**Mode (2), or inductor discharging mode:** During this mode, $Sb$ is switched OFF and the inductor discharges the energy through the inverter bridge antiparallel diodes to charge the capacitor, as shown in Figure 2b. The voltage $V_{inv}$ is equal to zero in this case and by applying the small ripple approximation, the equations describing the behavior of the inverter during this mode are as follows:

$$V_L = L\frac{di_L}{dt} = V_{dc} - V_c \tag{3}$$

$$V_{inv} = 0 \tag{4}$$

It is worth noting that during this mode, the inverter is disabled and accordingly, the average capacitor voltage is related to the boost duty cycle $D_b$ and can be expressed by Equation (5). After substituting $V_C$ from (5), $V_{inv}$ can be expressed in terms of $D_b$, as in Equation (6).

$$V_c = \frac{1}{1 - D_b} V_{dc} \tag{5}$$

$$V_{inv} = \frac{1}{1 - D_b} V_{dc} \tag{6}$$

### 2.2. The Operation of the Multi-Input CBBVSI Topology

The single-input CBBVSI can be extended into the multi-input topology, as shown in Figure 3. This topology can be used in distributed PV systems for microgrid applications. All the energy sources are PV modules and are controlled individually. The proposed inverter has the following advantages: (1) the power of each PV module can be harvested either separately or simultaneously, (2) the capability of single- or multi-input operation can be achieved, (3) a reduced number of components since only one switch and antiparallel diode for the inverter are used. The same operation for the single-input inverter, explained in the previous sections, applies for the proposed multi-input topology. It is worth noting that the modulation index is limited by the boost duty cycle. For the multi-input case, the maximum modulation index of the inverter depends on all boost duty cycles. This means that increasing the boosting ratio of any DC-DC converter increases the inverter modulation index. For example, if the minimum boosting ratio is equal to 5, the inverter modulation index is limited up to 0.8. On the other-hand, in **Mode (1)** all of the boost switches are turned ON, as shown in Figure 4a, and the capacitor voltage is applied to the inverter through the upper and lower switches. The boosting is achieved by switches ($S_{b1}$, $S_{b2}$, $S_{b3}$) since each inductor ($L_1$, $L_2$, $L_3$) charges from the corresponding PV input voltage throughout the corresponding boost switches. The inverter voltage $V_{inv}$ is equal to the capacitor voltage $V_C$. In **Mode (2)**, taking place when ($S_{b1}$, $S_{b2}$, $S_{b3}$) are turned OFF, the inductors ($L_1$, $L_2$, $L_3$), discharge the energy through the inverter bridge antiparallel diodes to charge the capacitor. The inverter voltage $V_{inv}$ is equal to zero, as shown in Figure 4b.

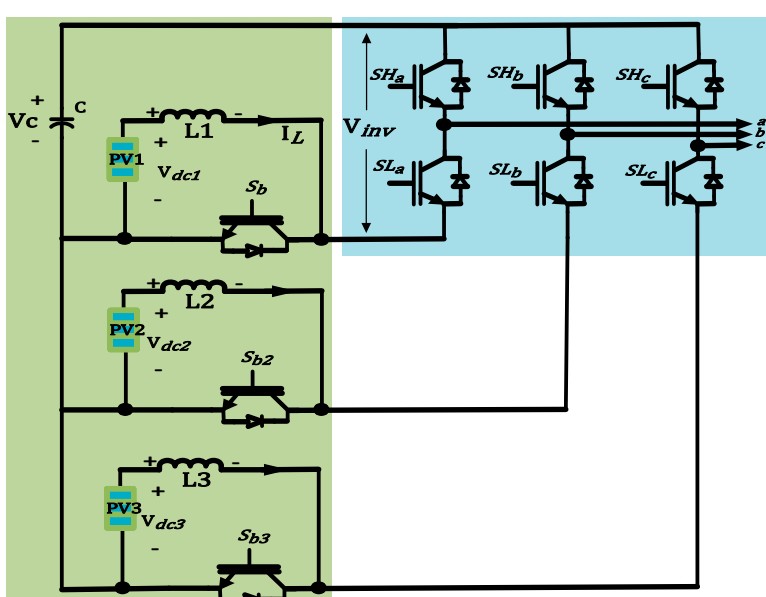

**Figure 3.** The proposed multi-input Ćuk-derived buck-boost inverter.

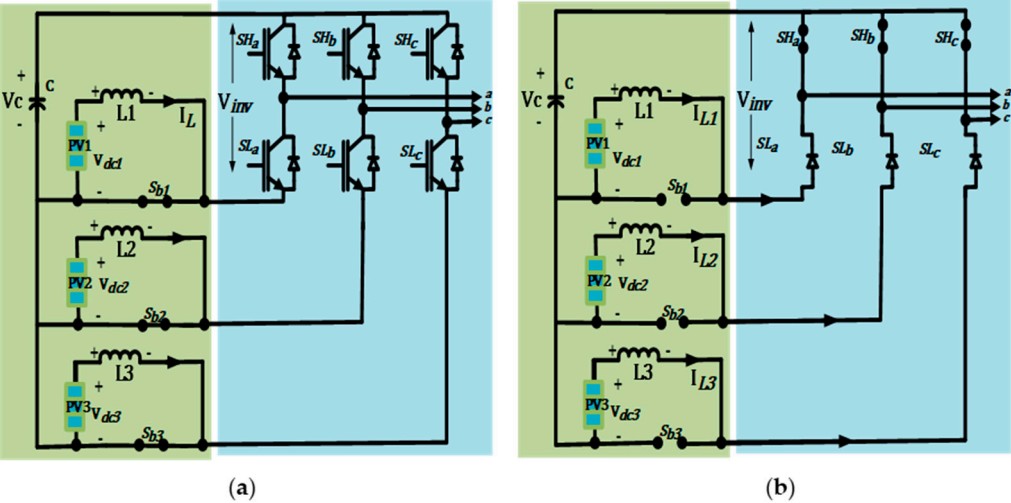

**Figure 4.** The simplified equivalent circuit diagram of the proposed multi-input inverter; (**a**) Mode 1: inductor charging; (**b**) Mode 2: inductor discharging.

In the case of one or two boost switches being turned OFF, the inductor corresponding to the switch that is switched OFF discharges the energy in the capacitor throughout the antiparallel diodes of the inverter leg. For the switch which is turned ON, the corresponding inductor charges the energy from the PV input. Meanwhile, all of the higher switches ($SH_a$, $SH_b$, $SH_c$) are turned ON and the lower switches ($SL_a$, $SL_b$, $SL_c$) are turned OFF to allow for the antiparallel diodes operation. Therefore, the voltage across the inverter is zero. The capacitor voltage is related to the minimum boost duty cycle applied to ($S_{b1}$, $S_{b2}$, $S_{b3}$) and can be expressed as in Equation (7) and $V_{inv}$, as in Equation (8).

$$V_c = \frac{1}{1 - D_{bst}} V_{dc} \tag{7}$$

$$V_{inv} = \frac{1}{1 - D_{bst}} V_{dc} \tag{8}$$

where $D_{bst}$ is the minimum duty cycle among all the boost switches and $V_{dc}$ is the corresponding input voltage for this minimum duty cycle.

## 3. Analysis of the Modulation of the Proposed Topology

### 3.1. Sinusoidal Pluse Width Modualtion

The proposed inverter is controlled by a sinusoidal pulse width modulation (SPWM) with a simple modification of the duty cycle $D_{bst}$. To create the required switching states, the switching signals for the full-bridge switches ($SH_{a,b,c}$, $SL_{a,b,c}$) are generated from the given buck and boost duty cycles $D_{bck}$ and $D_{bst}$, respectively. A sinusoidal reference waveform is compared with a high-frequency (fsw) triangular carrier to generate $D_{bck}$. The switches ($S_{b1}$, $S_{b2}$, $S_{b3}$) are controlled by the MPPT controllers with a switching frequency 2fsw to generate the duty cycle $D_{bst}$. As mentioned before, DC-AC conversion can be achieved when all switches ($S_{b1}$, $S_{b2}$, $S_{b3}$) are ON with duty cycle $D_{bst}$. Therefore, the switches ($SH_{a,b,c}$, $SL_{a,b,c}$) are controlled by the duty cycle $D_{bck}$ and $D_{bst}$. On the other hand, the states of the inverter are achieved as long as ($S_{b1}$, $S_{b2}$, $S_{b3}$) are ON with duty cycle $D_{bst}$. Otherwise, zero state is applied when one switch of ($S_{b1}$, $S_{b2}$, $S_{b3}$) is OFF. Figure 5 shows the simplest logic control for the higher (*SH*) and lower (*SL*) switches, and the required Boolean function for these switches can be summarized as follows:

$$S_L = \overline{D_{bck}}.D_{bst}$$

$$S_H = \overline{\overline{D_{D_{bck}}}.D_{bst}}$$

where $D_{bst} = min\{D_{b1}\ D_{b2}\ D_{b3}\}$. Figure 6 shows the switching pattern for one leg during one the line period.

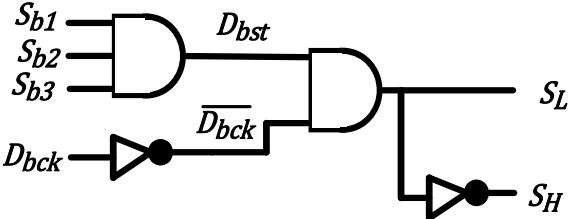

**Figure 5.** The simplest logic control for higher (*SH*) and lower (*SL*) switches.

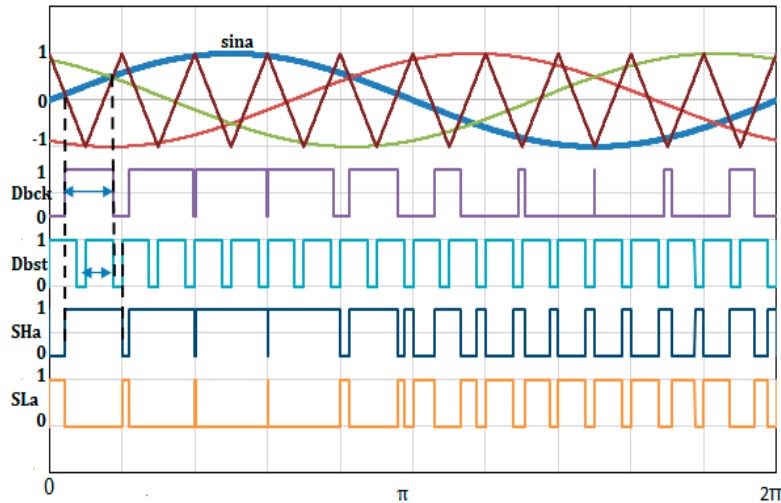

**Figure 6.** The switching pattern for the higher (*SH*) and lower (*SL*) switches and boost switch during one-line cycle.

*3.2. The Conversion Ratio*

The modulation index is defined as the relation between the phase voltage $V_{ph}$ and the inverter voltage $V_{inv}$ and can be expressed as follows:

$$M = \frac{V_{ph}}{Vinv/2} \tag{9}$$

From Equations (8) and (9), the relation between the peak phase voltage $V_{ph}$ and the DC input voltage is given by the following expression:

$$\frac{V_{ph}}{V_{dc}} = \frac{M}{2(1-D)} \tag{10}$$

where D is the duty cycle for the boost converter ($D_b$ in the case of a single-input operation). For a multi-input operation, it is the minimum duty cycle for all of the boost switches ($D_{bst}$).

From the operation description in Section 2, it is important to note that the maximum value of the modulation index coincides with the duty cycle of the boost DC-DC converters, i.e.,

$$M_{max} = D \tag{11}$$

The maximum conversion ratio G (the total gain) of the inverter is given by:

$$G = \frac{V_{ph}}{V_{dc}} = \frac{D}{2(1-D)} \tag{12}$$

The value of the operating duty cycle D is mainly determined by the MPPT controllers according to weather conditions and the imposed maximum power voltage V*mpp* and current I*mpp*. Finally, the current stress is summarized in Table 1, where the value of the maximum current is represented for every single switch. Due to the boosting operation, the RMS value of the input current I*in* is always larger than the RMS value of the phase current I*ph*. The input current I*in* is determined by the MPPT controller and is equal to the maximum power point (MPP) current I*mpp*. It is worth noting that the maximum current in each boost converter switch is the inductor current plus the phase current (I*ph* + I*in*) while the current of the other switches is equal to the phase current I*ph* for the proposed inverter.

**Table 1.** Devices stress.

| Symbol | The Maximum Current | The Maximum Voltage |
|:---:|:---:|:---:|
| $SH_{a,b,c}$, $SL_{a,b,c}$ | $I_{ph}$ | $V_{inv}$ |
| $S_{b1,2,3}$ | $I_L + I_{ph}$ | $V_{inv}$ |

## 4. Control of the Proposed Multi-Input Three Phase Inverter

The control system of the proposed energy conversion system consists of three blocks, as shown in Figure 7:

1. MPPT algorithm to harvest the maximum power from the PV sources. Any MPPT algorithm can be used for the proposed inverter. The waveforms of the PV voltage strongly depend on the type of MPPT controller used and the values of its parameters. If we restrict ourselves to a Perturb and Observe (P and O) MPPT controller, which is the one used in this paper, many variants of this algorithm exist. Namely, the output of the MPPT block could be a current reference that should be tracked by the PV current, a voltage reference that should be tracked by the PV voltage, or it could be the duty cycle that must be directly applied to the converter. In the first two cases, PI compensators are used to process the error between the controlled variables and their references. In the third case, the duty cycle is applied without any compensation scheme. Since the focus of the paper is not on this particular aspect, the simplest P and O MPPT controller providing the duty cycle directly is used in this paper. Its flowchart is shown in Figure 8. Each PV input has its own MPPT controller. For the proposed multi-input inverter, the output of this block determines the state of the corresponding boost switches ($S_{b1}$, $S_{b2}$, $S_{b3}$). Therefore, any change in the duty cycle of one of these boost switches, and the other two switches are not affected. Thus, the states are logically processed through a logic date of **AND** type to identify the maximum inverter modulation index, i.e., the inverter can be operated only if all the boost switches are ON.

2. The second control block is used to regulate the DC link voltage. The DC reference (V$c_{ref}$) should be determined based on the required boost voltage. A Proportional Integral (PI) controller is applied to this block. The Laplace domain transfer function of this controller can be expressed as follows:

$$G_{dc}(s) = K_{pdc} + \frac{K_i}{s} \tag{13}$$

The output of this block determines the reference current for the third control loop.

3. The third block controls the converter output current. The Proportional plus Resonant (PR) controller is adopted here. Its Laplace domain transfer function is given by the following expression:

$$G_P(s) = K_p + \frac{K_i}{s^2 + \omega^2} \tag{14}$$

where $\omega$ is the fundamental frequency.

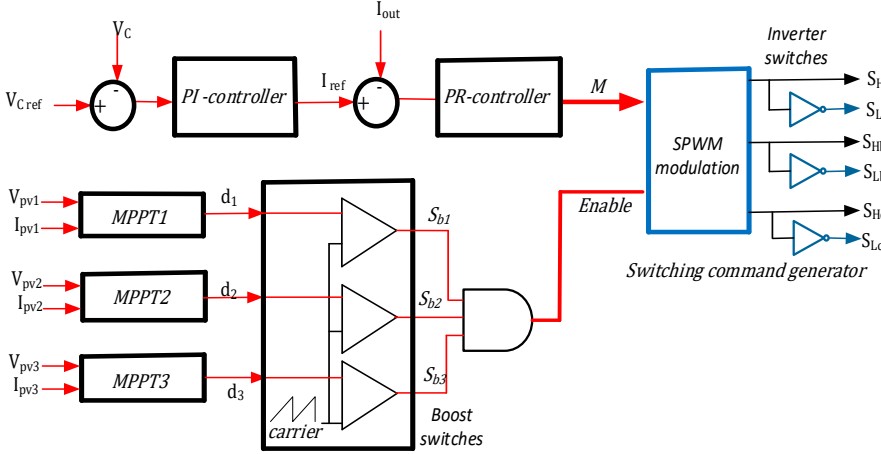

**Figure 7.** The control system of the proposed inverter.

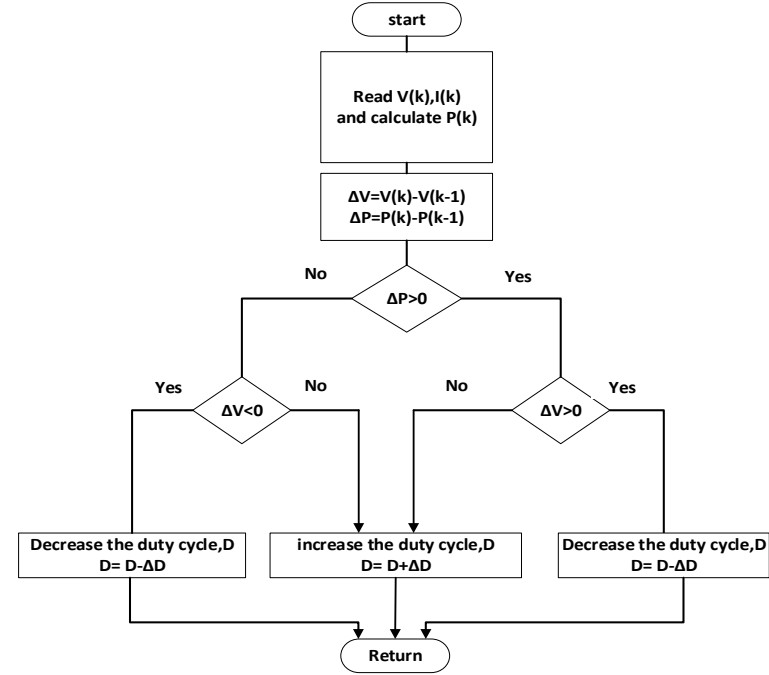

**Figure 8.** The flow chart of the Perturb and Observe (P and O) MPPT algorithm of each input operating independently.

The aforementioned control methods for the proposed single- and multi-input inverter are the same as the control methods adopted for other conventional converters. The other common controllers and MPPT methods can also be used.

## 5. Simulation Results

In order to verify the correct operation of the proposed inverter topology, the Matlab/simulink package is used. The DC inputs consist of a string of PV sources where three modules are combined in a series to form a single PV energy source. For a multi-input operation, each leg has an individual string of PV modules. The output power of the inverter is injected to a grid with an RMS voltage of 110 V and line frequency of 50 Hz. The used parameter values in the simulation are shown in Table 2. Different case studies are considered as follows:

**Table 2.** Simulation Parameter.

| Symbol | Quantity | Value |
|---|---|---|
| $V_{ph}$ | Grid voltage | 110 V RMS |
| $V_{oc}$ | Open circuit voltage of PV array | 112 V |
| $I_{sc}$ | Short circuit current of PV array | 7 A |
| $V_{max}$ | Voltage at MPP of PV array | 100 V |
| $I_{max}$ | Current at MPP of PV array | 6 A |
| $C$ | DC link capacitor | 47 uF |
| $L$ | Inductor $L_i$ of boost circuit | 1 mH |
| $F_{sw}$ | Switching frequency | 10 kHz |
| $K_p/K_i$ | PI and PR controller gains | 0.5/10 and 50/8000 |

**Single-input operation:** The grid phase voltage and current and DC-link voltage waveforms are illustrated in Figure 9a, while the power for each input is shown in Figure 9b for the single-input operation. From Figure 9, it can be observed that the current is in phase with voltage and that the capacitor voltage is regulated to its desired value of 380 V in steady-state, while the power is about 600 W.

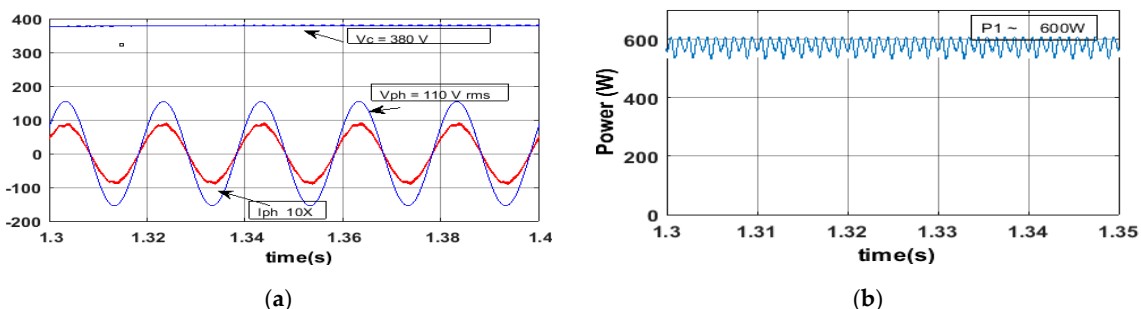

(a)                                                            (b)

**Figure 9.** Steady-state response from numerical simulation of the single-input operation (**a**) the capacitor voltage $V_C$, the grid phase voltage $V_{ph}$, and current $I_{ph}$, (**b**) the PV power for each input.

**Multi input operation:** The simulation results for the multi-input operation are shown in Figure 10, where it can be observed that the grid current is in phase with the phase voltage and the DC-link voltage is settled, also at the same value 380 V in steady-state operation, as shown in Figure 10a. The power for each input is 600 W with a total power of about 1.8 kW, as shown in Figure 10b. It is clear that more current is injected in the multi-input operation case due to the higher power resulting from the three input energy sources.

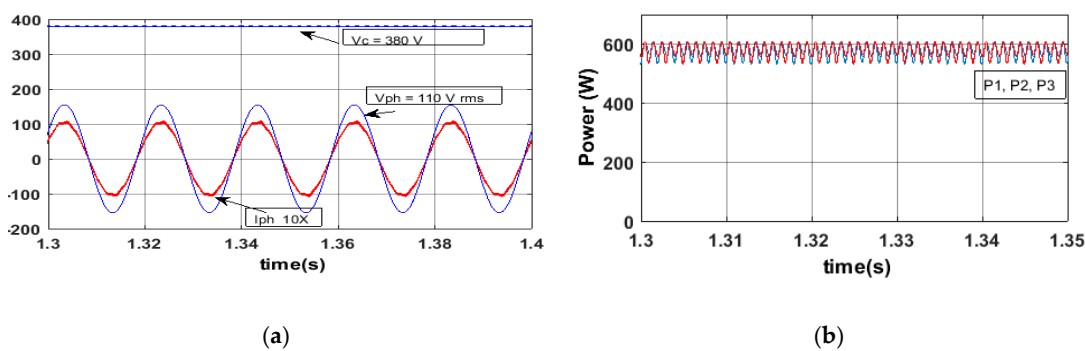

(a)                                                            (b)

**Figure 10.** Steady-state response from numerical simulation of the multi-input operation (**a**) the capacitor voltage $V_C$, the grid phase voltage $V_{ph}$, and current $I_{ph}$, (**b**) the PV power for each input.

**Multi input operation under partial shadow**: To simulate a partial shadow on one PV source, a change in the power of that source is applied at the time instant 0.945 s and removed at the time instant

1 s, as shown in Figure 11. It is clear from the waveforms depicted in Figure 11b that the change in one of the PV sources does not affect the extracted power from the other PV sources. The duty cycle of the shaded PV source ($d_3$) is changed to track the maximum power, as shown in Figure 11c, while the other duty cycles ($d_1$ and $d_2$) are not affected, as shown in Figure 11d,e. Note that the steady-state value of the capacitor voltage is the same as before and that a negligible overshoot is exhibited during the transient response.

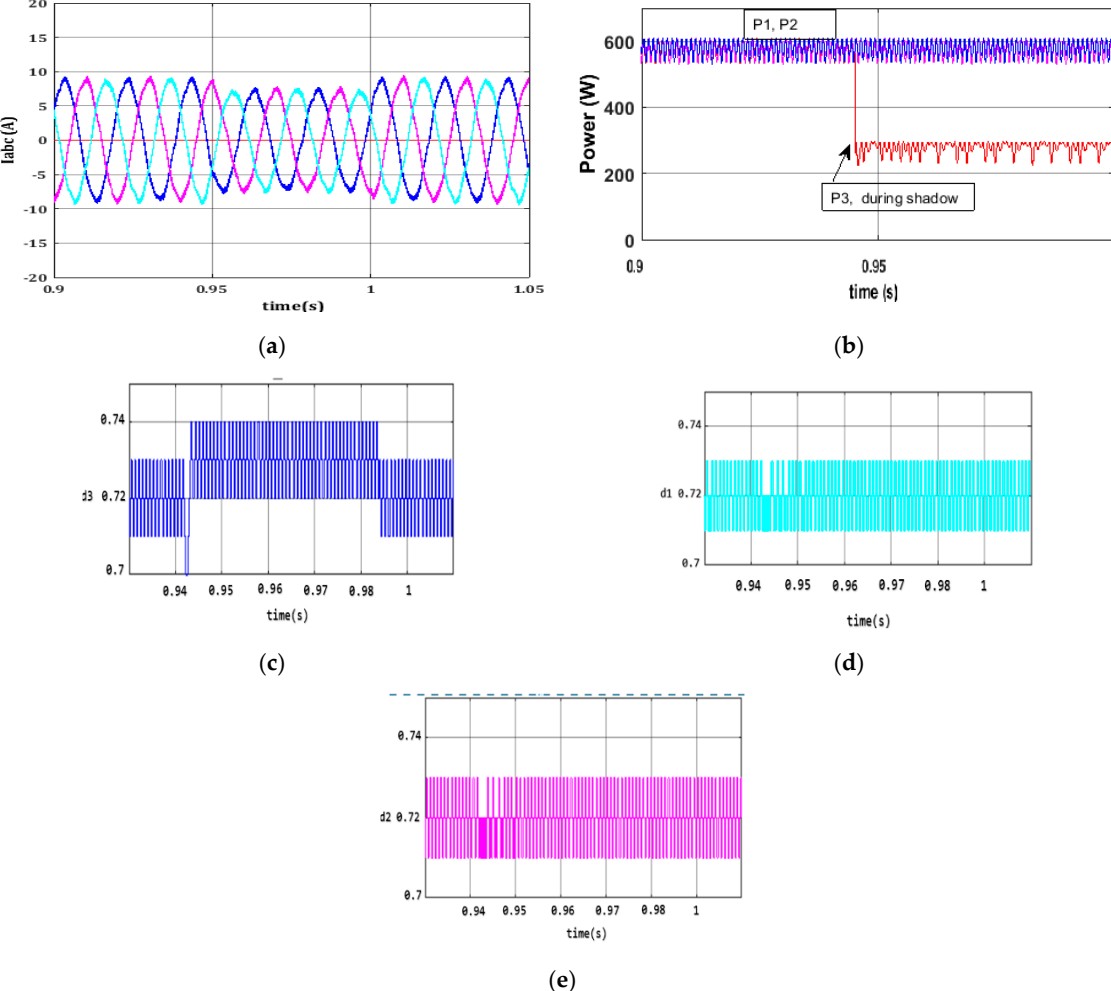

**Figure 11.** Simulation result of multi-input operation under partial shadow condition (**a**) the three-phase current, (**b**) the capacitor voltage $V_c$, and the three input power values, (**c**) the duty cycle $d_3$, for shadowed PV input, (**d**) the duty cycle $d_2$, during the shadow condition, (**e**) the duty cycle $d_1$, during the shadow condition.

## 6. Experimental Result

Using the values of system parameters listed in Table 3, the proposed inverter is constructed and connected to an RL load to verify the correct operation of the topology. The PV panel used was (JWP60 250 (www.jurawatt.de)) with the specifications listed in Table 3 (at standard test conditions, STC). The experimental setup is shown in Figure 12. The first test was conducted throughout the hourly PV power curve obtained in real measurements under the same conditions of the following tests with a peak power of 175 W, as depicted in Figure 13. It was reported in [29] in the same location that the maximum power in summer was less than 180 W at the peak hours due to the high-temperature and low-radiation. Three PV panels were series-connected to each inverter leg. The control scheme was implemented using the F28335 DSP board and FPGA board. Experimental tests have been carried

out at different operating conditions, namely, single-input, multi-input and under a partial shadow condition on some PV modules.

**Table 3.** Experimental parameters.

| Symbol | Quantity | Value |
|--------|----------|-------|
| $V_{oc}$ | Open circuit voltage of PV modules | $3 \times 37$ V |
| $I_{sc}$ | Short circuit current of PV modules | 9 A |
| $V_{max}$ | Voltage at MPP of PV module | $3 \times 30$ V |
| $I_{max}$ | Current at MPP of PV module | 8 A |
| *R load* | Resistor load (single-multi input) | 40–77 |
| *L filter* | Output current filter | 5 mH |
| C | DC link capacitor | 47 uF |
| L | Inductor $L_i$ of boost circuit | 1.2 mH |
| *fsw* | Switching frequency | 10 kHz |

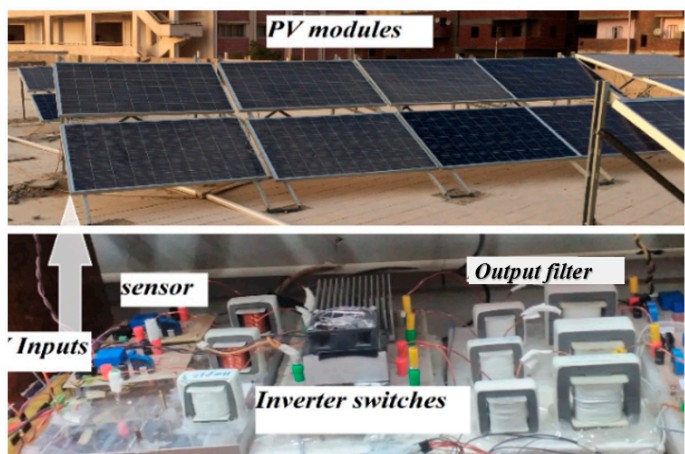

**Figure 12.** The experimental setup and PV modules.

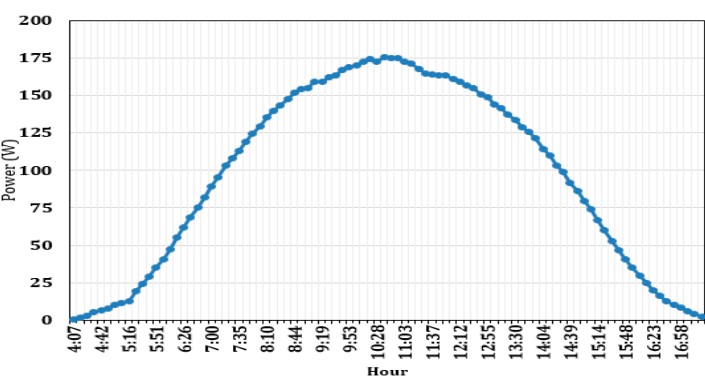

**Figure 13.** Power profile for one PV module.

**Single input operation:** In this test, three PV modules were connected in a series with the aim to increase the total voltage. The reference voltage of the DC link was set at 380 V. The PV voltage reached 86 V and the current was about 5.8 A, as shown in Figure 14a. The total input power was 498 W. The power was 166 W for each PV module. The load current and voltage and the DC link voltage ($V_c$) are shown in Figure 14b. From this figure, the output power was about 466 W with a conversion efficiency of about 93.6%.

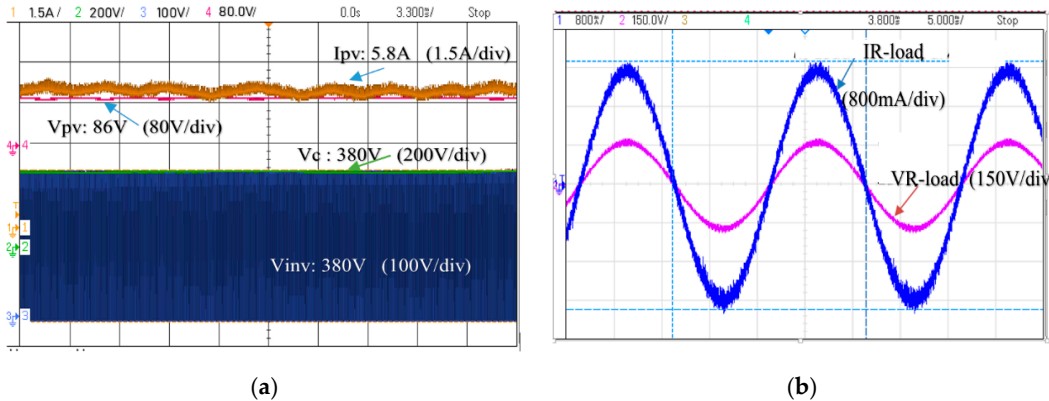

(**a**)          (**b**)

**Figure 14.** Experimental result for single-input operation: (**a**) PV current and voltage, the capacitor voltage $V_C$, and the inverter voltage $V_{inv}$, (**b**) the output current and voltage on the resistive load.

**Multi input operation:** In this test, each input was attached to three PV modules connected in a series. The reference voltage of the DC link $V_C$ was set at 380 V. The waveforms corresponding to this test are shown in Figures 15 and 16. The PV voltages and currents are depicted in Figure 15a,b, their average values being 83.5 V and 4 A, respectively. It can be observed that all PV sources work at their MPP with a total power of 1 kW. The output voltage and current and the DC link voltage $V_C$ are shown in Figure 16a. The output inverter levels are shown in Figure 16b, their peak value being 380 V, and was equal to the capacitor voltage $V_C$. The inverter efficiency was determined using the measurements at the input and load sides and was found to be approximately 92.4%.

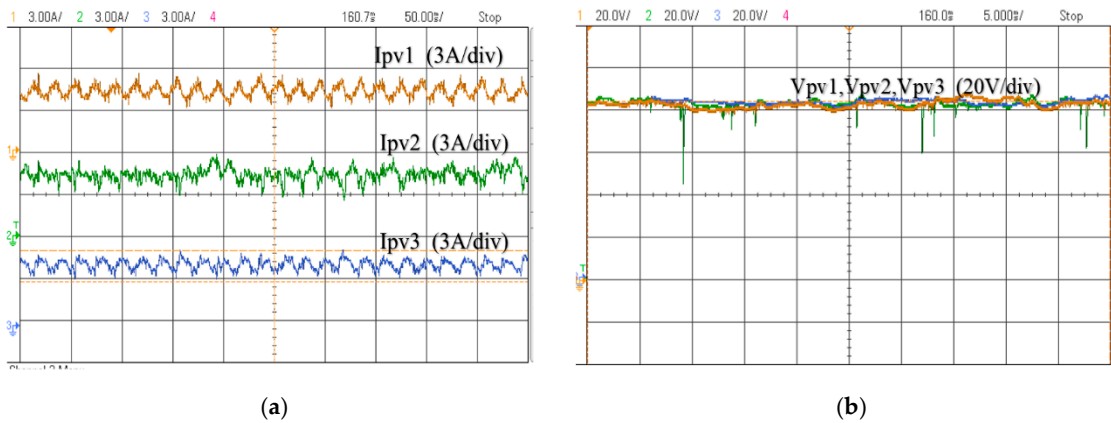

(**a**)          (**b**)

**Figure 15.** Experimental result for multi-input operation: (**a**) PV current, (**b**) PV voltage.

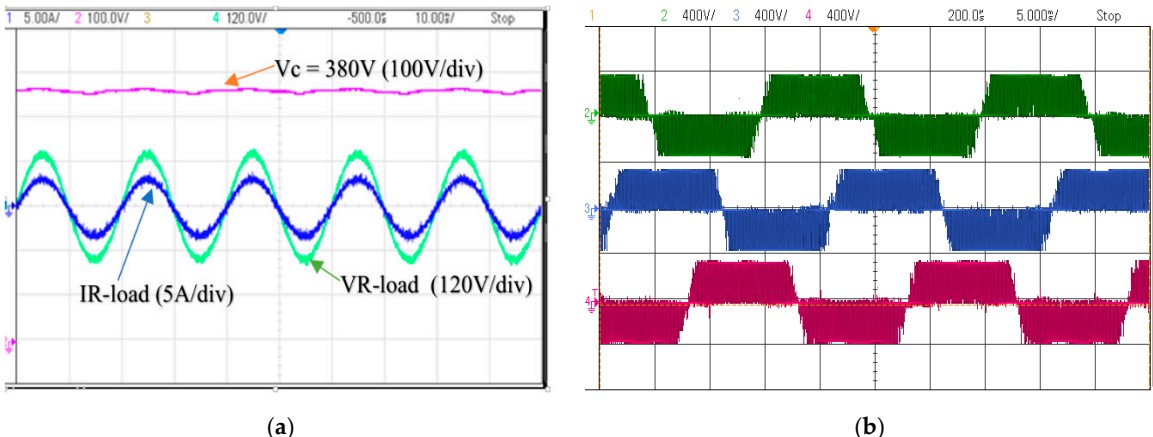

**Figure 16.** Experimental result for multi-input operation: (**a**) the capacitor voltage V$_C$, the output current and voltage on the resistive load, (**b**) the inverter voltage levels.

**The operation under partial shadow condition:** In this test, all the PV arrays were firstly connected to the inverter inputs. During the operation, one of the PV arrays was partially covered to emulate a partial shading condition and the results are depicted in Figure 17, where it can be observed (Figure 17a) that the PV current suddenly decreased. The load current settled at a new value due to the reduction of the PV source's power. The load current waveform corresponding to this test is shown in Figure 17b. The measured current was replotted using Matlab/Simulink and the result is shown in Figure 18. The total harmonic distortion (THD) of the output current was calculated and its value was 1.38%.

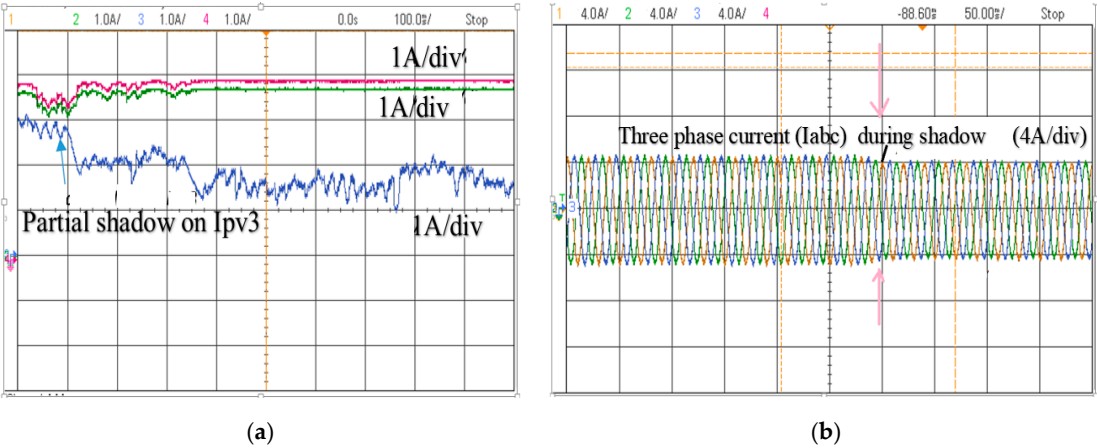

**Figure 17.** Experimental result for multi input operation during partial shadow: (**a**) PV current, (**b**) the output current.

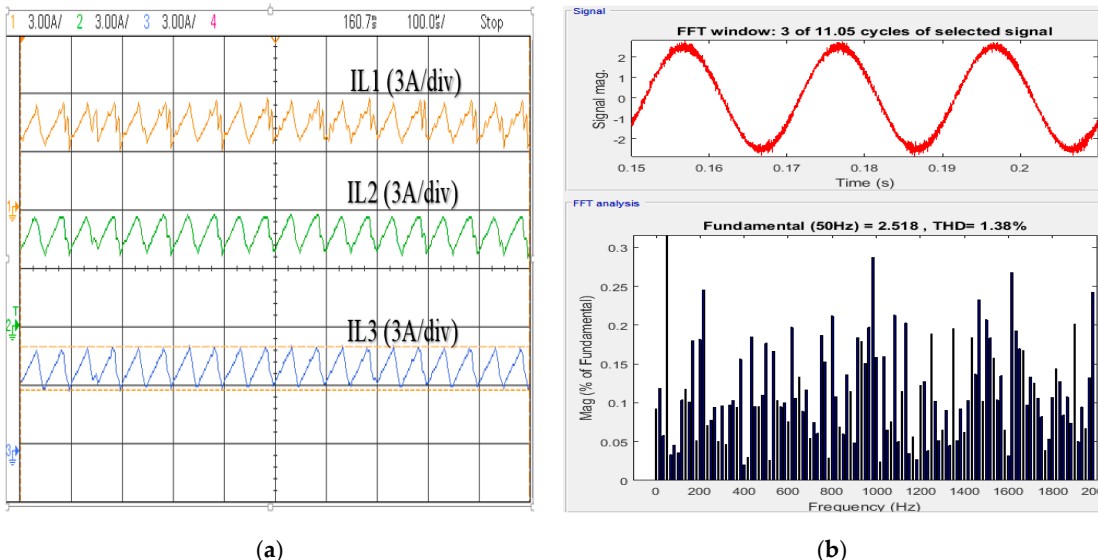

(**a**)　　　　　　　　　　　　　　　　(**b**)

**Figure 18.** Experimental result for (**a**) inductor current for each input, (**b**) load current and its harmonics contents showing a total harmonic distortion (THD) of about 1.38%.

## 7. Discussion

From the simulation and experimental results in the last two sections, it can be observed that there was a good match between the simulated and the measured waveforms of the main variables such as the capacitor voltage, load current, and voltage. For example, the simulated capacitor voltage is regulated to 380 V, as shown in Figures 9 and 10, and this is also the case for the experimental result depicted in Figures 14b and 16a. Similarly, under a shadow condition on the PV power for one PV module, the other two inputs are not affected, as shown in the simulation result depicted in Figure 11b. The same applies for the experimental result shown in Figure 17a.

In order to show the advantages of the proposed inverter, a number of selected topologies are compared, and the results of this comparison are summarized in Table 4 for both single- and multi-input topologies. For a fair comparison, the voltage gain considered is the ratio of the peak phase voltage to the input DC voltage. Also, the maximum measured efficiency for the selected topologies is considered. It is clear that the proposed inverter offers single-input operation with lower passive elements, higher voltage gain, and lower current stress compared to the topologies reported in [10], [12,21]. Compared to an SSI topology, replacement of three diodes with one MOSFETs (for single-input operation) is a clear advantage of the proposed inverter. The use of MOSFETs allows bi-directional power-flow for both inversion and rectification. In addition, higher efficiency can be achieved. As mentioned in the previous analysis, due to the boosting operation, the input current is always larger than the phase current. Therefore, the phase current has a negligible effect on the inverter switches. Compared to SSI and ZSI, the proposed inverter has lower current stress. Figure 19 shows the variation in $V_{ph}/V_{dc}$ versus $V_{inv}/V_{dc}$, illustrating the voltage stress at the same voltage gain $V_{ph}/V_{dc}$ for both SSI topology and the proposed inverter.

**Table 4.** Comparison among the performances of the proposed inverter and the existing inverters.

| Symbol | SSI [21] | ZSI [10,12] | The Proposed Inverter | | [26] Dual |
|---|---|---|---|---|---|
| | | | Single | Multi | |
| No. of input source | 1 | 1 | 1 | 3 | 2 |
| No. of switches | 6 | 6 | 7 | 9 | 8 |
| No. of diodes | 3 | 1 | 0 | 0 | 0 |
| No. of Inductors | 1 | 2 | 1 | 3 | 2 |
| No. of Capacitors | 1 | 2 | 1 | 1 | 1 |
| Capacitor value | 380 | 590 | 47 | 47 | Not reported |
| Measured voltage gain | 1.56 | 1.13 | 1.86 | 1.86 | 3.75 |
| Measured peak efficiency | Not reported | | 93% | 92.4% | Not reported |
| Maximum current stress for the inverter bridge | $3 \times (I_L + I_{ph})$ | $3 \times (2I_L/3 + I_{ph})$ | $3 \times (I_{ph})$ | $3 \times (I_{ph})$ | $3 \times (I_{ph})$ |
| | $3 \times (I_{ph})$ | $3 \times (2I_L/3 + I_{ph})$ | $3 \times (I_{ph})$ | $3 \times (I_{ph})$ | $3 \times (I_{ph})$ |
| Boost switch | - | - | $1 \times (I_L + I_{ph}))$ | $3 \times (I_L + I_{ph})$ | $2 \times (I_L + I_{ph}))$ |

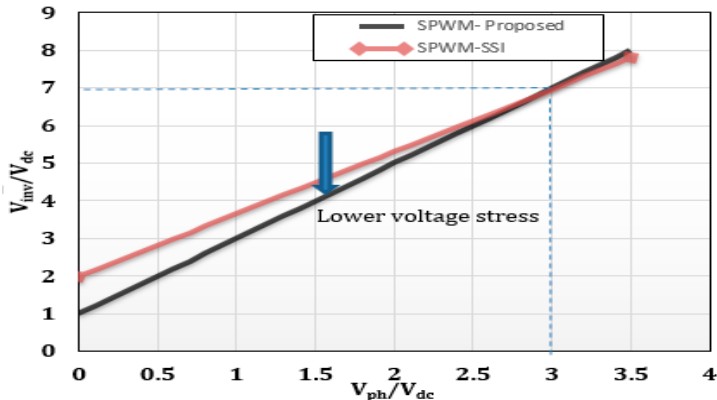

**Figure 19.** The conversion ratio $V_{inv}/V_{dc}$ versus $V_{ph}/V_{dc}$ for both Split Source Inverter (SSI) topology and the proposed inverter.

It is worth noting that with the proposed topology, a lower voltage stress can be achieved compared to SSI when a SPWM modulation is used with a gain $V_{ph}/V_{dc}$ less than three. For the distributed PV system, the proposed topology can be used for the single- and multi-input operation. Each input has its own MPPT controller and this reduces mismatch issues regarding shadow or environmental conditions. Regarding the size of the capacitor, with the proposed inverter a lower capacitance value can be used.

## 8. Conclusions

A Multi-input BBVSI inverter with distributed MPPT for microgrid applications was presented in this paper. The paper discussed the principle operation, and analysis, including the device stress and the voltage gain. Numerical simulations and experimental results of the proposed topology have been verified for different case studies and under diverse conditions. Different tests have been performed to validate the performance of the proposed inverter and its correct operation in this kind of application. One of the most important advantages of the proposed topology is the capability of operating with both single- and multi-inputs while using low-passive component values. This topology can be applied for grid-connected microgrid and also in standalone PV operations, in both cases reaching high efficiency and being of a relatively low cost. The main features of the proposed inverter are as follows: (1) a lower number of series modules are connected, reducing mismatch issues, and (2) multi-input of PV subarray can be operated at individual MPPT under shadowing conditions. The proposed inverter can

assure lower current stress for the switches compared to ZSI and SSI topologies. The response time and high efficiency MPPT are the most interesting issues in PV application, therefore future work will focus on developing MPPT algorithms with fast response times and higher efficiency.

**Author Contributions:** Conceptualization, E.A.; M.A.G.; E.M.A.; Methodology, M.O.; Software, E.A.; E.M.A.; Validation, M.A.G.; M.O.; Formal Analysis, M.O. and A.E.A.; Investigation, E.A.; Resources, M.O. and E.A.; Data Curation, E.A.; Writing—Original Draft Preparation, E.A.; Writing—Review & Editing, A.E.A. and M.O.; Visualization, E.A.; Supervision, M.O.; Funding Acquisition, M.O.

**Funding:** This work has been sponsored in part by the Egyptian Science and Technology Development Funds (STDF), project ID: 15261. A. El Aroudi would like to acknowledge the support of the Spanish Agencia Estatal de Investigación (AEI) and the Fondo Europeo de Desarrollo Regional (FEDER) under Grants DPI2017-84572-C2-1-R and DPI2016-80491-R (AEI/FEDER, UE).

**Conflicts of Interest:** The authors declare no conflict of interest.

## Abbreviations

The following abbreviations are used in this manuscript:

| | |
|---|---|
| CBBVSI | Ćuk-derived Buck-Boost voltage source inverter |
| MPPT | Maximum Power Point Tracking |
| PV | Photovoltaic |
| ZSI | Z-Source Inverter |
| qZSI | Quasi-Z-Source Inverter |
| SBBVSI | SEPIC-derived Buck-Boost voltage source inverter |
| SPWM | Sinusoidal Pulse Width Modulation |
| SSI | Split Source Inverter |

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
