# Peer review of "Multi-Input Ćuk-Derived Buck-Boost Voltage Source Inverter for Photovoltaic Systems in Microgrid Applications"

_energies, doi:10.3390/en12102007_

Round 1
Reviewer 1 Report
The paper is well organized, the objective is of high importance, and a proposed solution is useful. Good experimental results as well.

Author Response
Reviewer #1
The authors provide sufficient simulation and clear experimental results. However, it was written in Abstract: “The numerical simulations are presented together with experimental results showing a good agreement” – there wasn’t presented comparison between the simulation and the experiment. The comparison was between the proposed converter and the existed ZSI, and the propose converter did not prevail in all outcomes. I would suggest improving the last parts of the paper by detailed explanations of the simulation and experimental results.
Response: The authors would like to thank the reviewer for his valuable comments. The manuscript has been modified and a Discussion Section has been added providing detailed information about the simulation and experimental results.
From the numerical simulations and the experimental results in the previous sections, it can be observed a good matching between the waveforms of the capacitor voltage, load current and voltage. For example, the capacitor voltage is regulated to a peak voltage of 380 V in the numerical simulations as depicted in Fig. 9-10 and on the other hand in the experimental results shown in Fig. 14(b) and Fig. 16(a). Similarity, for the shadow effect on the PV power of one PV module, it can be observed that the other two inputs have no effect as shown in the numerical simulations shown in Fig. 11. The same apply to the experimental results shown in Fig. 17(a) .
Reviewer 2 Report
The figure shown in Fig. 1a does not appear to be the same with the figure in ref [23].
The rearranged figure of Fig. 1b is not the same as Fig. 1a. Can the authors explain how the re-arragnement proposed in Fig. 1b is the same with the other circuit?
The control of the proposed converter should be explained. How is control of the capacitor voltage achieved and which converter controls that voltage?
The proper function of the MPPT should be demonstrated. A reduction in the sun leads to lower power but how is actual operation at the MPP demonstrated in this work?
Is the grid frequency 50Hz or 60Hz?
The input power appears to vary with a second harmonic. This would not be the case in a converter with two stage tracking, or in fact any three phase converter. The authors should explain that operation of the proposed converter.
A complete diagram of the proposed control implementation should be provided. The gains for the controllers used should also be provided.
Under partial shadow, the currents of one module vary significantly, while the other two are almost constant. Can the authors explain this?
The inductor currents should be provided.
How is the dc voltage decided. Why is it 380V in one case and 370V in the second case?
A photo of the experiment must be provided.
In Fig. 14, the currents are not properly balanced in the three phases and the results are clipped. The figure must be revised.
The injected current from the FFT of Fig. 15 has 8% dc component. This is also seen in the simulation results and is not acceptable as most codes require a maximum dc current of 0.5%. This should be explained and addressed.
Table 2, the inductor value should be in mH, not mF.
Author Response
See attached PDF file

Reviewer 3 Report
See the attached review file

Author Response
Response to Reviewer #3
1. In the introductory section, the main objectives of the work should be more clearly formulated
in response to the critical analysis of the current research stage, going through the next path: the
disadvantages of the existing systems à the research opportunity à the novel approach. In the same idea, the authors should refer more broadly to their published studies in the field, and
highlight the elements that differentiate the current work from the previous works.
Response: Thanks for the reviewer for his comment and for providing interesting guidelines. The main objectives were formulated according to the provided path.
Regarding the published studies, a conference paper has been proposed on the performance analysis of single-input three-phase buck-boost inverter in [a]. However, a consistent performance evaluation of the Ćuk-derived BBVSI is still missing and requires and an investigation of the system behavior, especially, when operated as multi-input inverter. In addition, in our previous paper shown below presented in an IEEE conference, very few simulation results were presented without any experimental validation.
E. Abdeen, M. A. Gaafar and M. Orabi, "Performance Analysis for Single-Stage Buck-Boost Inverter," 2019 International Conference on Innovative Trends in Computer Engineering (ITCE), IEEE. Aswan, Egypt, 2019, pp. 587-592
One of the important points that is still required more research is the increase of the voltage gain on multi-input three-phase inverters, so single PV module can be used from obtaining required output grid voltage.
2. The research methodology must be more clearly described; a schematic representation of the
research workflow could clarify the approach.
Response: Thank you very much for your comment. The research methodology followed in our paper starts from mathematical modeling, analysis and design, numerical simulations and finally experimentally validating the results. This has been summarized in the introduction.
3. The quality of some figures should be improved, mainly in terms of text/notation clarity - size.
Response: Thank you very much for your suggestion. The quality of some figures has been improved as requested by the reviewer.
4. All the parameters used in the equations must be explained with the first use; check the entire
work in this regard.
Response: Thank you for your comment. The parameters for the equations have been illustrated and explained in the manuscript just when they are first used.
5. The information presented in the 2nd and 3rd sections of the work is partly known from the
scientific literature, so that they should be presented in a more synthetic form (in addition, the
two sections could be integrated into a single one).
Response: Thank you very much for your comment and suggestion. Section 2 mainly describes the basic operation of single input operation for clarity and simplicity of explanation. This is followed by the description of the operation of multi-input operation in Section 3. Sections 2 and 3 have been updated and merged in one section (Section 2) as requested by the reviewer.
6. Indicate the measures units for the variation diagrams in all the figures (some of them are
missing).
Response: Thank you very much for your suggestion. The measures units in all the figures have been indicated. See below:
7. The conclusions section should be extended and improved by a more detailed discussion on the research findings, and providing future research directions in the field, in terms of research
opportunities opened by this work.
Response: Thank you very much for your suggestion. A Discussion section were added where detailed information about the numerical simulation and experimental results is provided. Future Research was also proposed.
Round 2
Reviewer 2 Report
Thank you for the answers to the comments.
The derivation of the three-phase circuit from the original single boost converter is still not explained. The original converter shares the dc side of the system but in your topology each phase is connected independently. This variation is not explained in detail and is not trivial to the system.
Your experimental results still remain from an RL load and not from the grid connected system of the simulations.
Some more explanations on how the M = D limitation applies to the converter and what it means for the dc-side of the converter when connecting to the grid.
Can the authors provide the capacitor current and the voltages of the PV modules in the simulation results. I would also suggest presenting a figure with the individual PV curves to show tracking of the MPPT.
Can the authors show (even in the letter) the response of the converter for a change in the dc voltage reference, for example from 380V to 420V.
Is 10kHz the switching frequency of both the dc-dc converter and the inverter?
Round 3
Reviewer 2 Report
Thank you for the answers to the comments.
One minor question - given the variations in the PV voltage that you show in your current review, how exactly is MPPT achieved. These voltage waveforms differ substantially from what actual MPPT voltages look like in PV inverters.
Author Response
Thank you for the answers to the comments.
One minor question - given the variations in the PV voltage that you show in your current review, how exactly is MPPT achieved. These voltage waveforms differ substantially from what actual MPPT voltages look like in PV inverters.
Reply: Thank you very much for your efforts and comments to improve the readability of this paper.
We have added the following text to make clear how the used MPPT controller Works. Please see red text from line 203 to line 212.
"The waveforms of the PV voltage strongly depend on the type of the MPPT controller used and the values of its parameters. If we restrict ourselves to a P&O MPPT controller, which is the one used in this paper, many variants of this algorithm exist. Namely, the output of the MPPT block could be a current reference that should be tracked by the PV current, a voltage reference that should be tracked by the PV voltage or it could be the duty cycle that must be directly applied to the converter. In the first two cases, PI compensators are used to process the error between the controlled variables and their references. In the third case, the duty cycle is applied without any compensation scheme. Since the focus of the paper is not on this particular aspect, the simplest P&O MPPT controller providing directly the duty cycle is used in this paper. "